# Top-down modulation of olfactory-guided behaviours by the anterior olfactory nucleus pars medialis and ventral hippocampus

Afif J. Aqrabawi[1,*], Caleb J. Browne[2,*], Zahra Dargaei[1], Danielle Garand[1], C. Sahara Khademullah[1], Melanie A. Woodin[1] & Jun Chul Kim[1,2]

Olfactory processing is thought to be actively modulated by the top-down input from cortical regions, but the behavioural function of these signals remains unclear. Here we find that cortical feedback from the anterior olfactory nucleus pars medialis (mAON) bidirectionally modulates olfactory sensitivity and olfaction-dependent behaviours. To identify a limbic input that tunes this mAON switch, we further demonstrate that optogenetic stimulation of ventral hippocampal inputs to the mAON is sufficient to alter olfaction-dependent behaviours.

[1] Department of Cell and Systems Biology, University of Toronto, 25 Harbord Street, Toronto, Ontario, Canada M5S 3G5. [2] Department of Psychology, University of Toronto, 100 St. George Street, Toronto, Ontario, Canada M5S 3G3. * These authors contributed equally to this work. Correspondence and requests for materials should be addressed to J.C.K. (email: kim@psych.utoronto.ca).

The ability to adjust olfactory sensitivity is essential for animals to attend to and process the most relevant stimuli in a rapidly changing environment. Such coordination has been hypothesized to arise from centrifugal feedback originating from cortical structures such as the anterior olfactory nucleus (AON), piriform cortex and entorhinal cortex, which send direct excitatory inputs to the olfactory bulb (OB)[1–4]. Much attention has been paid to the feedforward stream of the olfactory pathway; however, owing to recent advances in methods that allow more selective manipulation of specific circuit elements, our understanding of olfaction is increasingly incorporating the functional contributions of cortical feedback[2,5,6].

The density of cortical projections to the OB illustrates the significance of centrifugal feedback to olfactory processing. In fact, cortical feedback projections to the OB outnumber olfactory sensory neuron inputs to the OB[3,6]. The largest source of cortical feedback projections to the OB originates in the AON, a ring-like cortical structure situated immediately caudal to the OB and rostral to the piriform cortex, which provides direct excitatory inputs to both inhibitory interneurons and mitral cells[1,7,8]. Despite the apparent importance of AON-derived cortical feedback in olfaction, the function of this input has not been directly demonstrated in awake behaving animals, leaving its relevance unknown[9,10]. Here, we demonstrate that a subdivision of the AON, the pars medialis (mAON) bidirectionally controls olfactory sensitivity and olfaction-dependent behaviours, and identify a limbic input from the ventral hippocampus (vHPC) that is capable of tuning mAON activity.

## Results

**mAON inhibition enhances olfaction-dependent behaviours.** To examine the behavioural function of cortical feedback inputs to the OB, we virally expressed the chemogenetic activity silencer hM4D bilaterally in CaMKIIa-positive neurons of the mAON. The mAON is unique within the olfactory system[10–12], projecting heavily to the ipsilateral OB, but lacking projections to other downstream olfactory cortical areas[13,14]. Therefore, inhibiting mAON activity can selectively eliminate a major portion of cortical feedback to the OB without affecting olfactory processing in other cortical areas. hM4D expression was restricted to the mAON and did not spread to the piriform cortex or other subregions of the AON (Fig. 1a,b). Confocal imaging of the OB revealed axon terminals of mCherry-containing mAON neurons primarily in the deep granule cell layer with a few scattered fibres present in the glomerular layer, as previously described[14,15] (Fig. 1b). In line with previous work[16], whole-cell patch-clamp recordings confirmed that bath application of clozapine-N-oxide (CNO), the synthetic ligand for hM4D, hyperpolarized the membrane potential and inhibited current injection-evoked action potential firing in hM4D-expressing neurons (Supplementary Fig. 1).

On the basis of its direct, excitatory influence on OB granule cells[1], it is possible that the mAON is involved in gating incoming olfactory signals. Thus, we first examined whether inhibition of mAON feedback could alter the threshold for odour detection. Mice were presented with cotton swabs containing increasing concentrations of a highly diluted odour (from 0.001 to 0.1% odour concentration), and time spent investigating the swab was measured as a behavioural parameter for an animals' ability to detect the odour[17]. CNO treatment did not alter the baseline investigation of mineral oil (0% odour concentration; Fig. 1c, inset) but enabled detection of the odour at a low concentration (0.001% odour concentration; Fig. 1c) compared with vehicle treatment, suggesting that inhibition of feedback projections from the mAON enhanced olfactory detection sensitivity. Notably,

mAON inhibition did not alter the ability of animals to detect, or habituate to, novel odours when presented in high concentrations (Supplementary Fig. 2a).

We then examined whether this enhanced olfactory sensitivity altered performance in two olfaction-dependent behaviours: foraging in the buried food test and social interaction, both critically dependent on olfaction in mice[18–20]. CNO treatment facilitated the performance of food-deprived mice in uncovering a chocolate reward buried under a deep layer of cage bedding (Fig. 1d). This effect cannot be explained by an increase in movement, as CNO treatment had no effect on locomotor activity measured in an open field (Supplementary Fig. 2b). Social interaction was measured in separate sociability and social recognition phases. CNO treatment increased the proportion of time spent investigating a stranger conspecific compared with an empty cage, indicative of an increased sociability. Subsequently, in the social recognition phase, CNO treatment increased the proportion of time spent interacting with a novel mouse compared with a previously introduced mouse, indicative of an increased ability to differentiate between novel and familiar conspecifics (Fig. 1e). Importantly, in a sham surgery group, CNO treatment did not produce any effect on these olfaction-dependent behaviours (Supplementary Fig. 3).

**mAON activation impairs olfaction-dependent behaviours.** We next determined whether mAON activation was sufficient to change the threshold of odour detection. In a separate group of mice, the chemogenetic activator hM3D was expressed in CaMKIIa-positive neurons of the mAON. Similarly to hM4D experiments, hM3D expression was largely restricted to the mAON (Fig. 2a). Whole-cell patch-clamp recordings confirmed that application of CNO depolarized the membrane potential and increased current injection-evoked action potential firing in hM3D-expressing neurons (Supplementary Fig. 1). CNO treatment did not alter baseline investigation of mineral oil (0% odour concentration; Fig. 2b, inset) but impaired detection of a diluted odour (0.001, 0.01, and 0.1% odour concentration; Fig. 2b), suggesting that activation of feedback projections from the mAON reduced olfactory detection sensitivity. Nonetheless, mAON activation did not change animals investigation of, and habituation to, novel odours of a high concentration (Supplementary Fig. 4a). In the buried food test, activation of the mAON substantially impaired the ability of food-deprived mice to uncover a buried food reward (Fig. 2c). In the social interaction test, mAON activation impaired social recognition as mice treated with CNO spent an approximately equal proportion of time investigating a novel and familiar conspecific (Fig. 2d), supporting the role of the mAON in modulating olfactory sensitivity. However, unlike mAON inhibition, activation of the structure produced no change in investigation time during the sociability phase, when a stranger conspecific is present only in one side of the arena, while the other side is left empty. It is likely that despite their reduced olfactory sensitivity, mice relied on other sensory (for example, visual) stimuli to guide their investigation of the individual conspecific. Collectively, these results demonstrate a distinct role for the mAON in the bidirectional control of olfactory sensitivity and olfaction-dependent behaviours.

While the data suggest that the observed changes in olfaction-dependent behaviours result from altered olfactory processing, these changes can be interpreted as indirect consequences of affecting other limbic areas involved in anxiety, exploration or motivation. To address this possibility, we tested new groups of mice expressing hM4D or hM3D in the mAON for their anxiety-like behaviour (elevated plus maze (EPM) and novelty-suppressed feeding test), general exploratory behaviour (hole-board test), and

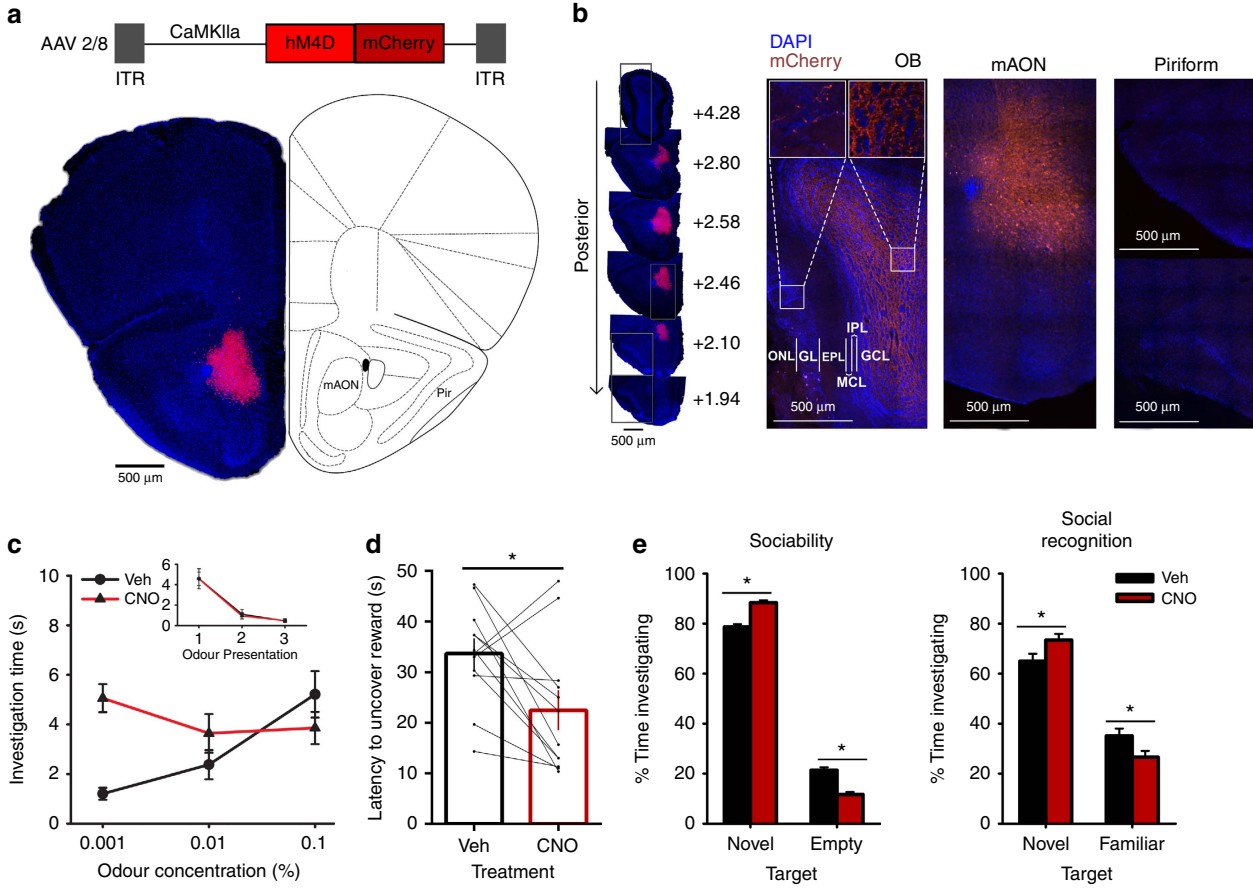

**Figure 1 | Inhibition of the mAON enhances olfactory sensitivity and the performance of olfaction-dependent behaviours.** (**a**) AAV-mediated expression of hM4D-mCherry in CaMKIIa-positive neurons was restricted to the mAON. ITR, inverted terminal repeats. (**b**, left) Serial sections depicting the extent of hM4D-mCherry expression at viral infusion site with AP axis coordinates from bregma for reference. (right) Confocal images of OB coronal sections depicting DAPI-stained nuclei and mCherry-positive axon fibres arriving from mAON. Right panels correspond to the boxed regions in the left panel. mCherry-positive axon terminals of mAON CaMKIIa-positive neurons were found to innervate primarily the deep granule cell layer with a few scattered fibres present in the glomerular layer of the OB (EPL, external plexiform layer; GCL, granule cell layer; GL, glomerular layer; IPL, internal plexiform layer; MCL, mitral cell layer; ONL, olfactory nerve layer), but virtually none are present in the anterior (above) or posterior piriform cortex (below). (**c**) CNO treatment did not alter investigation of mineral oil (0% odour concentration) across habituation trials (inset; data obtained from olfactory habituation/dishabituation test, Supplementary Fig. 2a), but increased investigation time of an odour at a low concentration compared with vehicle treatment ($n = 11$, two-way ANOVA interaction $F_{(2,20)} = 10.94$, *$P < 0.001$). (**d**) CNO treatment decreased the latency to locate a buried food reward ($n = 12$, paired-samples $t$-test, $t_{(11)} = 2.85$, *$P < 0.05$). (**e**) Compared with vehicle-treated mice, CNO-treated mice spent proportionally more time investigating a conspecific versus an empty cage ($n = 11$ per group, independent-samples $t$-test, $t_{(20)} = 6.32$, *$P < 0.001$; absolute investigation time for novel conspecific between groups, $t_{(20)} = 2.15$, *$P < 0.05$), and showed a higher preference for investigating a novel versus familiar conspecific ($t_{(20)} = 2.18$, *$P < 0.05$; absolute investigation time for novel conspecific between groups, $t_{(20)} = 6.27$, *$P < 0.05$). ANOVA, analysis of variance.

motivation for food reward (runway test). To demonstrate the preservation of the mAON's ability to alter olfaction-dependent behaviour in this new group of mice, we confirmed that manipulating mAON activity altered performance during the buried food test (Supplementary Fig. 5). In the EPM, mice expressing hM4D or hM3D displayed similar open arm entries and spent an equivalent percentage of time in the open arms following either CNO or vehicle treatment (Fig. 3a). In the novelty-suppressed feeding test, CNO treatment had no influence on the latency for hM4D or hM3D-expressing mice to begin feeding (Fig. 3b) nor did the treatment cause a change in the amount of food consumed in their home cage (Fig. 3c). In the hole-board arena, hM4D or hM3D-expressing mice displayed a similar number of head dips and head dip duration as well as number of rears and rearing duration following CNO or vehicle injections (Fig. 3d). Both hM4D and hM3D-expressing mice exhibited no difference in their latency to retrieve a food reward at the end of a metre-long runway following CNO or vehicle

treatment, indicating mAON manipulation had no effect on motivation to feed (Fig. 3e). Together, these findings demonstrate that changes in mAON function altered olfactory sensitivity and olfaction-dependent behaviours without affecting anxiety, general exploration or motivational measures.

**Activating vHPC inputs to mAON alters olfaction-dependent behaviours.** Output of the mAON is channelled into the OB, where it can bidirectionally modulate olfactory sensitivity and olfaction-dependent behaviours. Rather than working in isolation, the mAON likely operates as part of a broader cortical network supervising olfactory processing by receiving ascending limbic inputs from higher cortical structures. To identify a potential source of cortical input to the mAON, we conducted a retrograde tracing experiment. Infusions of the retrograde tracer cholera toxin B subunit into the mAON produced dense labelling in the ipsilateral vHPC CA1 and subiculum (Supplementary

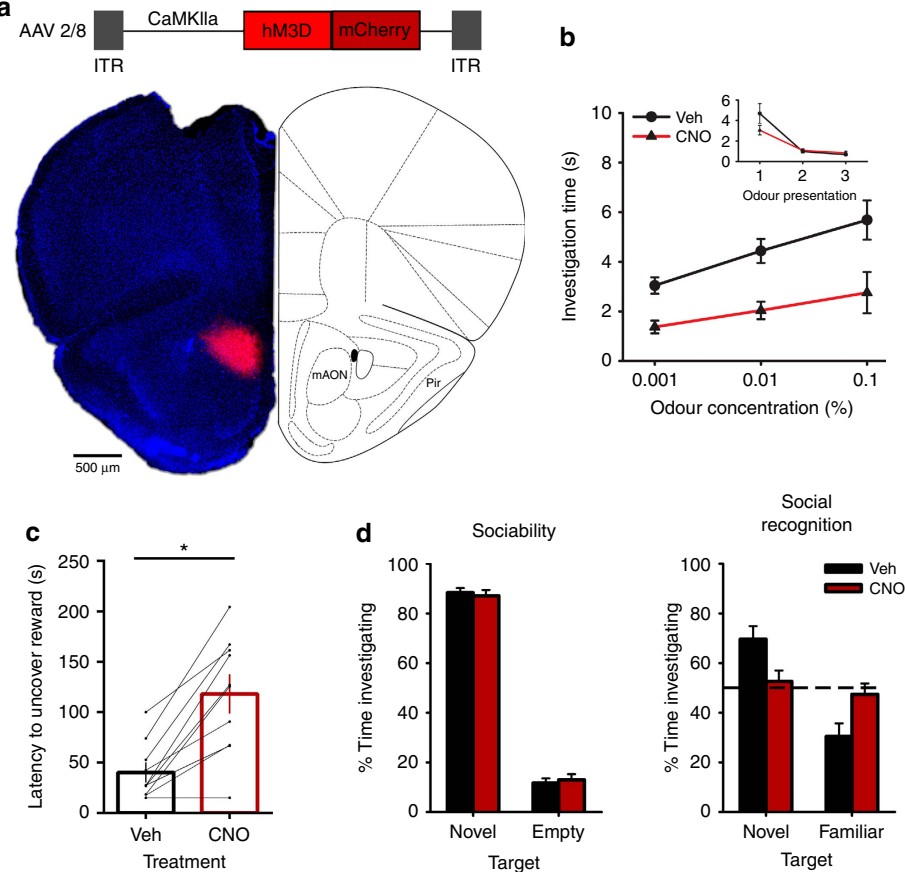

**Figure 2 | Activation of the mAON reduces olfactory sensitivity and impairs the performance of olfaction-dependent behaviours.** (**a**) AAV-mediated expression of hM3D-mCherry in CaMKIIa-positive neurons was restricted to the mAON. (**b**) CNO treatment did not alter investigation of mineral oil (0% odour concentration) across habituation trials (inset; data obtained from olfactory habituation/dishabituation test, Supplementary Fig. 4a), but impaired detection of an odour at a low concentration compared with vehicle treatment ($n = 12$, two-way ANOVA both main effects $F_{(2,22)} > 10.50$, $P < 0.001$). (**c**) CNO treatment increased the latency to locate a buried food reward ($n = 10$, paired-samples $t$-test, $t_{(9)} = 5.57$ *$P < 0.001$). (**d**) CNO-treated mice exhibited no change in sociability (vehicle group $n = 10$, CNO group $n = 11$, independent-samples $t$-test, $t_{(19)} = 0.41$, ns, $P = 0.68$), but failed to distinguish between a novel and familiar conspecific (independent-samples $t_{(19)} = 2.50$, $P < 0.05$; paired-samples $t$-tests, vehicle group $t_{(9)} = 3.71$, $P < 0.01$, CNO group $t_{(10)} = 0.60$, ns, $P = 0.56$; absolute investigation time for novel conspecific between groups, $t_{(19)} = 1.13$, ns, $P = 0.25$). ANOVA, analysis of variance.

Fig. 6), demonstrating a monosynaptic vHPC-mAON pathway. This finding is consistent with previous studies, which also theorize that the mAON relays limbic inputs from the vHPC onto the OB to adjust olfactory behaviours[13,21,22].

To test the function of vHPC inputs to the mAON, we used an optogenetic strategy wherein vHPC neurons were virally transduced with channelrhodopsin-2 fused to YFP (ChR2-YFP) or GFP alone (Fig. 4a and Supplementary Fig. 7). Strikingly, ChR2-YFP-positive vHPC terminals were tightly restricted to the mAON with little signal observed in the OB, piriform cortex or other AON subregions (Fig. 4a and Supplementary Fig. 8). Patch-clamp electrophysiology recordings revealed that photoactivation of vHPC terminals reliably elicited a response in the majority of mAON pyramidal neurons examined (Fig. 4b). These findings suggest that the vHPC has the potential to indirectly influence bottom-up olfactory processes through inputs to the mAON.

Selectively activating the vHPC-mAON pathway increased the latency of mice to uncover a buried food reward, with latency returning to baseline when tested again without stimulation (Fig. 4c and Supplementary Fig. 9). In the social interaction test, ChR2-YFP and GFP groups showed similar levels of sociability when no stimulation was applied. However, vHPC-mAON pathway stimulation abolished social recognition when a second novel conspecific was added to the arena (Fig. 4d). In a real-time

place preference test, mice spent equal amounts of time in a laser-paired and laser-unpaired chamber, indicating that activation of the vHPC-mAON pathway is neither aversive nor appetitive (Supplementary Fig. 10). The vHPC also projects to the prefrontal cortex (PFC) which is situated dorsal to the mAON. The density of ChR2-YFP-positive vHPC fibres arriving in the PFC are comparable to those terminating in the mAON (see Supplementary Fig. 8). It is possible that the behavioural effects observed following stimulation of the vHPC-mAON pathway could be the result of light loss along the length of the fibre, inadvertently activating the PFC. Activation of the vHPC itself and thus downstream targets aside from the mAON is also possible due to backward propagation of the photocurrent-induced action potential. To eliminate these possibilities, a separate group of mice that received bilateral infusions of ChR2-YFP or GFP to the vHPC and bilateral optical fibre implantations targeting the PFC were tested using identical photostimulation parameters to those in the vHPC-mAON experiments. Activation of vHPC fibres in the PFC did not alter the ability of mice to locate a buried food reward or distinguish between novel and familiar conspecifics in the social interaction test (Supplementary Fig. 11). Together, these findings demonstrate that optogenetic stimulation of vHPC inputs to the mAON is sufficient to alter olfaction-dependent behaviours.

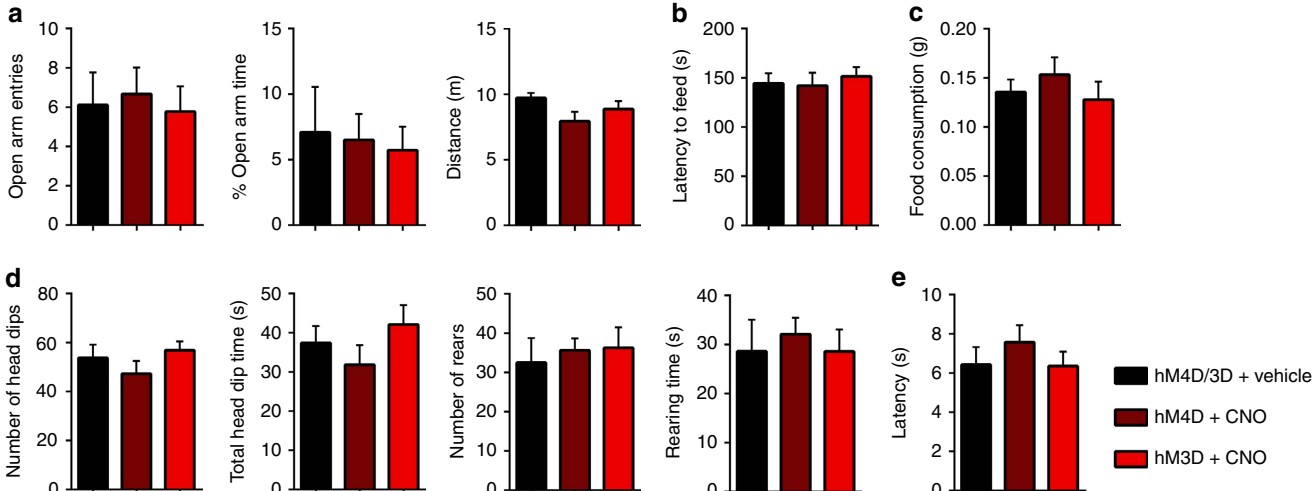

**Figure 3 | Modulation of mAON activity has no overt effect on affective behaviours.** hM3D-expressing mice and hM4D-expressing mice were each divided into two groups and treated with either vehicle or CNO before the behavioural tests. Vehicle-treated hM3D mice and hM4D mice showed no difference and, therefore, were combined. (**a**) In an elevated-plus maze, CNO-treatment in hM4D or hM3D-expressing animals did not influence entries into the open arms (left: independent-samples $t$-test; vehicle ($n = 9$) versus hM4D ($n = 9$), $t_{(16)} = 0.26$, ns, $P = 0.80$; vehicle versus hM3D ($n = 9$), $t_{(16)} = 0.16$, ns, $P = 0.88$), percentage of time spent in the open arms (middle: vehicle versus hM4D, $t_{(16)} = 0.39$, ns, $P = 0.70$; vehicle versus hM3D, $t_{(16)} = 0.62$, ns, $P = 0.54$), or total distance travelled (right: vehicle versus hM4D $t_{(16)} = 1.85$, ns, $P = 0.084$; vehicle versus hM3D, $t_{(16)} = 1.19$, ns, $P = 0.25$). (**b**) CNO-treated groups displayed similar latency to retrieve and consume food placed in the centre of a brightly lit open field compared with vehicle-treated groups (vehicle ($n = 9$) versus hM4D ($n = 9$), $t_{(16)} = 0.14$, ns, $P = 0.89$; vehicle versus hM3D ($n = 9$), $t_{(16)} = 0.52$, ns, $P = 0.61$). (**c**) CNO-treatment did not change the amount of food consumed in the home cage (vehicle ($n = 9$) versus hM4D ($n = 9$), $t_{(16)} = 0.82$, ns, $P = 0.43$; vehicle versus hM3D ($n = 9$), $t_{(16)} = 0.35$, ns, $P = 0.73$). (**d**) Exploratory behaviour in the hole-board test was not affected by inhibition or activation of the mAON as determined by measuring number of head dips (left: vehicle ($n = 8$) versus hM4D ($n = 8$), $t_{(14)} = 0.87$, ns, $P = 0.40$; vehicle versus hM3D ($n = 8$), $t_{(14)} = 0.48$, ns, $P = 0.64$), total head dip time (middle-left: vehicle versus hM4D, $t_{(14)} = 0.85$, ns, $P = 0.41$; vehicle versus hM3D, $t_{(14)} = 0.71$, ns, $P = 0.49$), number of rears (middle-right: vehicle versus hM4D, $t_{(14)} = 0.45$, ns, $P = 0.66$; vehicle versus hM3D, $t_{(14)} = 0.46$, ns, $P = 0.65$), and right: total rearing time (vehicle versus hM4D, $t_{(14)} = 0.48$, ns, $P = 0.64$; vehicle versus hM3D, $t_{(14)} = 0.0056$, ns, $P = 0.996$). (**e**) CNO-treated hM4D and hM3D groups exhibited no difference in latency to obtain food on a runway compared with vehicle-treated group (vehicle ($n = 8$) versus hM4D ($n = 8$), $t_{(14)} = 0.92$, ns, $P = 0.37$; vehicle versus hM3D ($n = 8$), $t_{(14)} = 0.064$, ns, $P = 0.95$).

## Discussion

Targeted manipulation of cortical feedback pathways in awake behaving animals is integral to understanding the contribution of top-down control in shaping olfaction-dependent behaviours. The present results suggest that mAON activity suppresses bottom-up olfactory processes, consistent with electrophysiological evidence that glutamatergic AON inputs to granule cells of the OB reduce local mitral/tufted cell activity[1,23]. At the behavioural level, inhibition of the mAON enhanced the detection of a weak odour, while odour detection was reduced following mAON activation. These results suggest that the mAON plays an inhibitory role in odour detection sensitivity. This is further supported by the ability of mAON inhibition to enhance both the localization of a buried food reward and recognition of a novel conspecific, while the opposite pattern of effects is produced by mAON activation. However, the mAON does not seem to be important when odour detection is easy: mAON inhibition or activation has no effect on the detection of, or habituation to, novel odours of a high concentration. Thus, the mAON appears to adjust olfactory sensitivity for detection of weak olfactory stimuli.

Robust cortical projections arising from the mAON to the OB granule cell layer suggest that cortical feedback inputs to the OB serve as a global gain control mechanism of olfactory signals by tuning the response magnitude of OB output neurons. Specifically, mAON activation is thought to increase inhibitory inputs to the mitral/tufted cells, lowering the net signal gain in the OB, whereas suppression of mAON activity produces an opposite effect. Notably, recent findings have shown that the

direct manipulation of granule cells in the OB changes odour discrimination ability[24,25]. Deletion of the AMPA receptor subunit GluA2 in granule cells or direct optogenetic activation of granule cells improved discrimination of similar odour mixtures, while conditional knockout of NMDA receptors or hM4D-mediated inhibition of granule cells impaired odour discrimination learning. These findings suggest that OB granule cells decorrelate or disambiguate overlapping neural representations of similar olfactory stimuli in the OB, thereby enhancing overall olfactory specificity[25].

The present results together with previous findings collectively point to a hypothesis that enhancing olfactory sensitivity (for example, by inhibiting OB granule cells) facilitates odour detection, but likely compromises olfactory specificity by introducing more noise to the neural representation of odour signals at the level of the OB. Accordingly, enhancing olfactory specificity (for example, by activating OB granule cells) would come at a cost of lowering olfactory sensitivity for odour detection. The balance between olfactory sensitivity and specificity may change dynamically depending on the task goal. For example, when the goal is to detect and locate a weak olfactory stimulus, it is beneficial to maximize the overall olfactory sensitivity by sacrificing olfactory specificity. Consistent with this idea, we found that activation of the mAON impaired the ability to detect the presence of an odour at low concentrations and to locate a buried food reward. However, when the goal shifts to discriminating similar olfactory stimuli against a complex odour background, it would become more beneficial to lose sensitivity and gain specificity. Such a trade-off

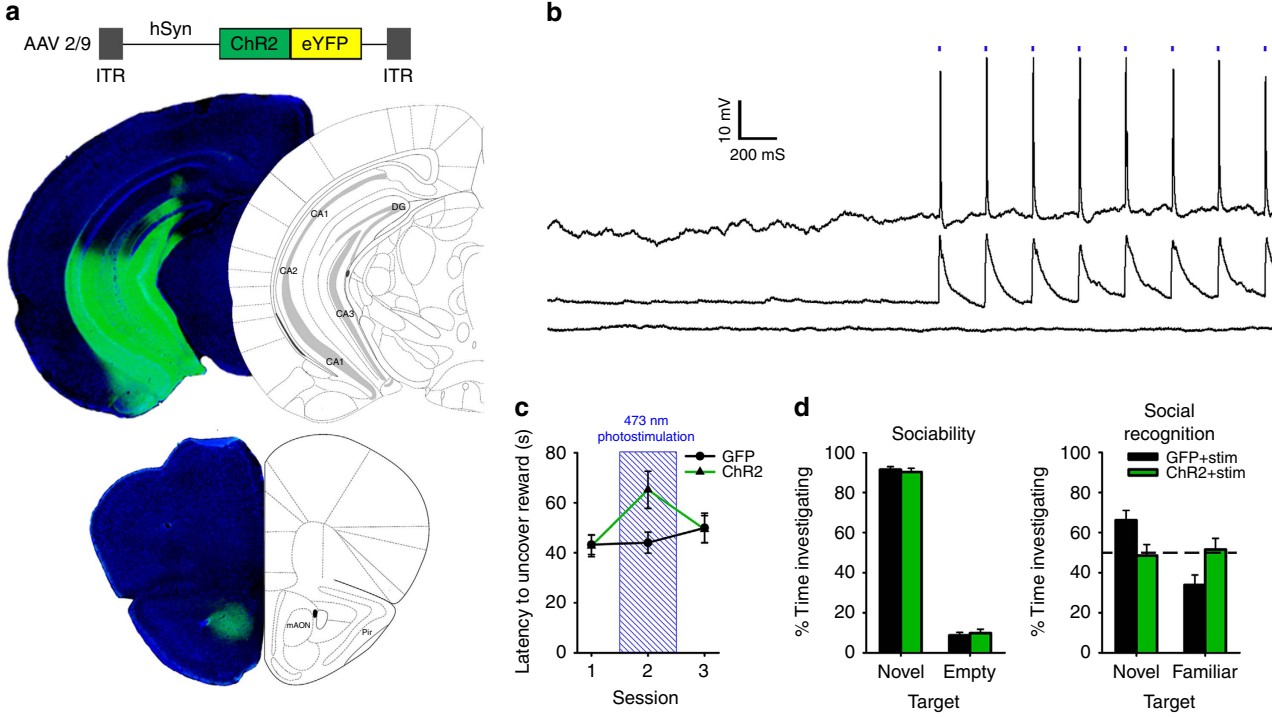

**Figure 4 | vHPC input to the mAON impairs olfaction-dependent behaviours.** (**a**) AAV-mediated expression of ChR2-YFP in the vHPC driven by the human synapsin promoter (hSyn) produced dense terminal fields localized to the mAON. (**b**) Example traces from *in vitro* current-clamped mAON neurons in response to stimulation of ChR2-containing vHPC terminals (5 ms pulses, 4 Hz, 5 mW). Cells showed reliable spiking activity ($n = 1$) or EPSPs ($n = 6$) following light pulses, while some exhibited no response ($n = 2$). (**c**) *In vivo* stimulation of the vHPC-mAON pathway (5 ms pulses, 4 Hz, 1 mW) significantly increased the latency to locate a buried food reward compared with control mice ($n = 8$ per group, two-way ANOVA interaction $F_{(2,28)} = 3.60$, $P < 0.05$; session 2, $t_{(14)} = 2.47$, $P < 0.05$). (**d**) In the absence of photostimulation, ChR2 and GFP mice showed no difference in the proportion of time spent investigating a novel conspecific versus an empty cage (left: ChR2 $n = 7$, GFP $n = 8$; independent-samples $t$-test, $t_{(13)} = 0.48$, $P = 0.64$). However, photostimulation impaired the ability of ChR2 mice to distinguish between a novel and familiar conspecific (right: independent-samples $t$-test, $t_{(13)} = 2.38$, $P < 0.05$; paired-samples $t$-tests, GFP group $t_{(7)} = 3.25$, $P < 0.01$, ChR2 group $t_{(6)} = 0.60$, ns; absolute investigation time for novel conspecific between groups, $t_{(13)} = 0.28$, ns). ANOVA, analysis of variance.

can enhance contrast between similar odours by suppressing overlapping neural representation evoked by similar odours while maintaining their non-overlapping neural activation patterns.

It should be noted that the current study did not directly examine olfactory specificity. Thus, future work will be needed to test how mAON activation and inhibition would impact olfactory discrimination in behavioural tests that require the segregation of a target odour from the overlapping background odours[26]. Building on our hypothesis, we predict that mAON activation will reduce the extent of overlap between the mitral cell responses to a target and background odours, improving overall olfactory discrimination. Intriguingly, we found that mAON activation did not enhance, but impaired, social recognition, while mAON inhibition improved social recognition. Social recognition is a complex, multi-step psychological process that likely requires both olfactory sensitivity and specificity. We speculate that performance in a social recognition task might be highly sensitive to changes in odour detection, such that the effects of enhanced or reduced olfactory sensitivity to detect social odours may have masked any potential effects of changing olfactory specificity.

It is possible that mAON projections to regions other than the OB can explain some of the behavioural effects observed in these experiments. For example, projections to the hypothalamus may alter food-related behaviours and affect performance of the buried food test (BFT), while projections to the amygdala may promote anxious states that disrupt behavioural performance in general[27]. However, manipulation of mAON activity had no effect on home cage feeding, motivation to acquire food or measures of anxiety. These results indicate that the effects of mAON manipulation on latency to uncover a buried food reward and social recognition are likely a result of changes in olfactory processes, specifically.

Dynamic modulation of olfactory sensitivity (and specificity) may be particularly adaptive for animals facing threatening or anxiety-provoking situations that necessitate extraction of information from the environment to determine an optimal course of action. Olfactory sensitivity may be directly tuned by limbic inputs from the vHPC, which has been implicated in a variety of emotional and cognitive processes such as anxiety, fear and memory[28,29]. It is thought that the vHPC contributes to such processes by monitoring the novelty and aversiveness of the environment and sending contextual information to its downstream cortical and subcortical targets. Here, we found that optogenetic activation of vHPC inputs to the mAON increased the latency of mice to uncover a buried food reward and impaired social recognition, indicative of decreased olfactory sensitivity. These findings are consistent with the effects of direct activation of the mAON, and suggest that the vHPC can serve as a source of excitatory input to the mAON to suppress olfaction-dependent behaviours. Thus, it is plausible that mAON receives information regarding the animal's current and previously experienced emotional states from the vHPC and integrates those inputs to adjust bottom-up olfactory processes accordingly. While the present study suggests that the vHPC is a candidate region for higher-order modulation of olfactory sensitivity via the mAON, future studies should identify the functional significance of other

neocortical and limbic structures positioned to alter olfaction through the mAON.

## Methods

**Animals.** Male C57BL/6N mice (Charles River) were used in all behavioural experiments and *in vitro* patch-clamp slice recordings. All were 8–10 weeks old at the beginning of experimental procedures. Before surgery, mice were group housed in a temperature-controlled room on a 12 h light/dark cycle with *ad libitum* access to food and water, unless otherwise specified. All procedures were performed in accordance with the guidelines of the Canadian Council on Animal Care (CCAC) and the University of Toronto Animal Care Committee.

**Surgical procedures.** AAV2/8-CaMKIIa-hM4D(Gi)-mCherry (hM4D) and AAV2/8-CaMKIIa-hM3D(Gq)-mCherry (hM3D) viral vectors were purchased from the Vector Core at the University of North Carolina, AAV2/8-hSyn-ChR2-eYFP (ChR2) and AAV2/8-CB7-CI-EGFP-RBG (GFP control) from the Vector Core at the University of Pennsylvania. Stereotaxic surgery was conducted on mice maintained on isoflurane anaesthesia. For hM3D and hM4D experiments, viral vectors were bilaterally infused into the mAON (10° angle towards midline targeting A/P: +2.5 mm, M/L: ±0.5 mm D/V: −3.5 mm in Paxinos and Franklin, 2007) in a volume of 0.1–0.2 μl. For retrograde tracing experiments, Alexa Fluor 488-conjugated CTB was infused into the mAON in a volume of 0.1 μl. For ChR2 experiments, viral vectors were bilaterally infused into the vHPC (10° angle away from midline, targeting A/P: −3.1 mm, D/V: −4.0 mm, M/L: ±3.25 mm) in a volume of 0.3–0.4 μl, and optical fibres (200 μm core, 0.39 NA; Thorlabs, Newton, NJ, USA) threaded through 1.25 mm-wide zirconia ferrules (Thorlabs) were bilaterally implanted directly above the mAON or PFC (10° angle towards midline targeting A/P: +1.5 mm, M/L: ±0.5 mm D/V: −2.25 mm). All infusions were made at a rate of 0.1 μl min$^{-1}$ by pressure ejection by means of a cannula connected by Tygon tubing to a 10 μl Hamilton syringe mounted on an infusion pump. Following surgery, animals were single housed and testing began 2 weeks later.

**Drugs.** CNO (obtained from the NIH as part of the Rapid Access to Investigative Drug Program funded by the NINDS) was dissolved in a solution of 10% DMSO and 0.9% saline. CNO or its vehicle was administered intraperitoneally in a volume of 10 ml kg$^{-1}$ 10 min before the start of all behavioural testing. A dose of 5 mg kg$^{-1}$ CNO was used throughout testing in all groups of animals. All drug treatments were randomized and separated by 72 h.

**Apparatus for optogenetic experiments.** Stimulation of vHPC terminals in the mAON was achieved by illumination with 473 nm blue light generated by a diode-pumped solid state laser (Laserglow, Toronto, ON, Canada), the output of which was controlled by a waveform generator (Keysight technologies, California, USA). For behavioural experiments, the laser was connected to a $1 \times 2$ optical commutator (Doric Lenses, Quebec, QC, Canada) which split the light beam into two separate arena patch cables that were attached to implanted optical fibres.

**In vitro patch-clamp slice recordings.** Brains were rapidly removed after decapitation and placed into a cutting solution containing the following (in mM): 205 sucrose, 2.5 KCl, 1.25 NaH2PO4, 25 NaHCO3, 25 glucose, 0.4 ascorbic acid, 1 CaCl2, 2 MgCl2 and 3 sodium pyruvate, pH 7.4, osmolality 300 mOsm kg$^{-1}$. Transverse slices (350 μm) containing the mAON were prepared from mice previously infected with AAV containing hM4D- or hM3D-mCherry in the mAON, or ChR2-GFP in the vHPC. Slices recovered at 32 °C in a 50:50 mixture composed of cutting saline artificial cerebrospinal fluid (aCSF) for 30 min and then placed in aCSF alone for 30 min. During experimentation slices were perfused at a rate of 2 ml min$^{-1}$ in aCSF. The aCSF solution consisted of the following (in mM): 123 NaCl, 2.5 KCl, 1.25 NaH2PO4, 25 NaHCO3, 25 glucose, 2 CaCl2 and 1 MgCl2 in double-distilled water and saturated with 95% O2/5% CO2, pH 7.4, osmolarity 300 mOsm. Whole-cell patch-clamp recordings were obtained from transduced neurons in the mAON. Micropipettes were filled with an intracellular fluid containing the following (in mM): 130 potassium gluconate, 10 KCl, 10 HEPES, 0.2 EGTA, 4 ATP, 0.3 GTP and 10 phosphocreatine, pH 7.4, osmolality 300 mOsm kg$^{-1}$.

For hM4D and hM3D experiments, neurons were identified by mCherry fluorescence. The resting membrane potential was recorded (in current-clamp mode) in normal aCSF for a minimum of 5 min before the addition of 1 μM CNO for 10 min. Neuronal excitability was determined in current-clamp mode by injecting current in 10 pA steps from 50 to 150 pA.

For the current-induced action potential analysis in hM3D and hM4D samples, all recordings were performed on minimum of three animals per group. Recordings were initiated 10 min after membrane rupture. A series of 500 ms currents at increasing steps (100–300 pA) were injected to neurons, and the number of action potentials was quantified in the neurons firing multiple action potentials.

For ChR2 experiments, whole-cell patch-clamp recordings were obtained from putative pyramidal neurons in the mAON. Membrane potential changes were measured while 5 ms pulses of 473 nm blue light were applied (5 mW, 4 Hz).

**Olfactory sensitivity test.** In a 10 min habituation session, mice were placed in a clear plexiglass test cage measuring $12 \times 16 \times 12$ cm, which contained a Q-tip soaked with mineral oil or water. The arena was enclosed and the Q-tip was inserted through a hole in the roof. Following habituation, 7.5 μl of increasing concentrations (0.001, 0.01 and 0.1%) of peanut butter dissolved in mineral oil or almond extract dissolved in distilled water were presented to mice on the Q-tip. The amount of time spent directly investigating the Q-tip was examined in consecutive 5 min trials each separated by a 5 min inter-trial interval. Each trial was recorded at a rate of 60 fps using a NIKON D5200 video camera equipped with a 30 mm prime lens. Scoring was completed offline using frame-by-frame analysis from the side-view video of the chamber, and the criterion for scoring investigation was head up sniffing within 1 cm of the Q-tip. Chewing or direct manipulation of the Q-tip was not scored. In accordance with a within-subjects design, animals were tested twice following treatment with either CNO or its vehicle in a counterbalanced fashion.

**Habituation/dishabituation test.** Testing was carried out in the same chamber as that used in olfactory sensitivity. Mice were first habituated to the test arena containing a dry Q-tip for 10 min. In 3 min trials, mice were sequentially presented with new Q-tips permeated with water, two non-social neutral odours, or two social odours, each presented for three trials with a 2 min inter-trial interval. The amount of time spent investigating the Q-tip on each trial was scored. The criterion for scoring investigation was head up sniffing within 1 cm of the Q-tip. Chewing or direct manipulation of the Q-tip was not scored. Habituation was defined as a reduction in investigation of a repeatedly presented odour, while dishabituation was defined as a reinstatement of investigation of the Q-tip when a novel odour was presented. The non-social odours used include almond extract, vanilla extract, banana extract and mineral oil. Social odours were obtained from swabs of bedding from two cages of three adult conspecific males and two cages of three conspecific females, which had not been changed for ~1 week. Investigation time following CNO or its vehicle was always assessed using a counterbalanced within-subjects design. As such, in the first session, mice were presented with odours of water, almond, vanilla, male cage 1 and female cage 1. In the second session, mice were presented with odours of water, banana, mineral oil, male cage 2 and female cage 2.

**Buried food test.** Two days before behavioural testing, mice were acclimatized to the food reward (M&M mini) by placing two samples in their home cage overnight. Experiments were performed in a $40 \times 12 \times 12$ cm clear plexiglass chamber filled with 3 cm of corncob bedding. Mice received three 10 min habituation sessions on three consecutive days wherein one piece of food reward was placed on top of the bedding in the centre of the arena. Animals were then overnight food restricted before testing. In a test session, mice were given three consecutive trials to find $\frac{1}{4}$ of an M&M mini (0.06 g) pseudorandomly buried beneath the bedding in one of four approximate locations (four corners or centre). Immediately after being placed in the arena, the latency to find the buried food was measured using a stopwatch; the trial ended when mice visibly retrieved and began consuming the food. Within a test session, each mouse received three consecutive trials to find the buried food and the average of these three trials were taken as the latency score. Between trials, the bedding was replaced and the chamber wiped with 70% ethanol solution. Mice first underwent three baseline testing sessions (that is, without CNO or vehicle injections), after which the scored test following the experimental manipulation began. Animals were tested twice, after treatment with either CNO or its vehicle, in a counterbalanced fashion. Following each test session, mice were given *ad libitum* access to chow and testing was carried out every other day to allow for adequate food consumption following restriction. In the vHPC-mAON ChR2 condition, mice were tested once with optical stimulation of vHPC terminals in the mAON (4 Hz 1 mW, 5 ms pulse width) throughout each trial, and again without stimulation. For the vHPC-PFC condition, mice were only tested once with stimulation applied during each trial. Two GFP control mice from the vHPC-PFC condition were excluded from data analysis due to a consistent lack of engagement in testing.

**Social interaction test.** Social interaction was assessed using Crawley's three-chamber apparatus, consisting of three 40 cm wide × 20 cm long × 40 cm high chambers. Removable partitions (5 cm wide × 40 cm high) in the walls allowed animals to traverse between chambers. The two far chambers each contained a cylindrical wire cage with a diameter of 10.5 cm and height of 11 cm with bars spaced 1 cm apart, while the middle chamber was empty. Testing was conducted under red light and the presence of tone generators to minimize reliance on visual and auditory cues. Each session involved three separate 10-min phases: habituation, sociability and social recognition. Each phase was separated by a two minute interval. Test animals were always placed in the centre chamber and removal of both partitions marked the beginning of the test. In the habituation phase, animals were allowed to acclimate to the entire test arena with partitions removed and both wire cages empty. In the sociability phase, a young adult male C57BL/6 mouse unfamiliar to the test subject was added to a wire cage on one side of the arena (counterbalanced between subjects), while the other wire cage was left empty. In the social recognition phase, a second juvenile male mouse, also unfamiliar with the test subject, was added to the wire cage that was previously

occupied and the first, now-familiar, animal was placed in the previously empty cage (counterbalanced between subjects). In both sociability and novelty preference phases, behaviour was recorded using an overhead camera with ANY-maze (Stoelting). Direct interactions were manually scored, offline, from overhead videos. Criteria used for scoring were direct contact of subject's nose with the wire cage or mouse contained therein, while climbing on the wire cage was excluded from scoring. Data were presented as a percentage time spent investigating the target cage over the entire time spent engaged in investigation of either cages. All mice were tested once in accordance with a between-subjects design following treatment with either CNO or its vehicle. In our hands, repetition of this test in the same animal was not possible; data between first and second test sessions was inconsistent between test animals. For ChR2 experiments, mice underwent 10 min habituation and sociability phases wherein no stimulation was applied, followed by a 7 min social recognition phase during which optical stimulation (4 Hz 1 mW, 5 ms pulse width) of vHPC terminals in the mAON or PFC was applied throughout.

**Open field test.** Mice were placed in a clear Plexiglas box measuring 50 cm wide × 50 cm long × 20 cm high. Distance travelled was measured in a 30 min session from an overhead video using ANY-maze (Stoelting) Animals were tested twice following treatment with either CNO or its vehicle in a counterbalanced fashion.

**Real-time place preference.** Mice in the ChR2 condition were placed in an arena with two distinct chambers measuring 10 × 20 × 15 cm. One chamber had vertical stripes on the walls and the other with an array of filled circles. To engage olfactory circuits, each chamber also had an enclosed petri dish with five holes drilled into the top containing the spices ginger (vertical stripes) or thyme (circles). Entry of the animal into one chamber was paired with constant laser stimulation (4 Hz, 1 mW, 5 ms pulse width) controlled by an input/output signal from predefined regions of an overhead video using ANY-maze (Stoelting). The laser-paired chamber was counterbalanced between animals. Mice could move freely between chambers and time spent in each chamber was measured in a 10 min session.

**Elevated plus maze.** The EPM apparatus consists of two black plastic open arms (30 cm × 5 cm) perpendicularly conjoined at a centre (5 cm × 5 cm) with two plastic enclosed arms (30 cm × 5 cm × 30 cm). The maze was elevated one meter from the ground. In a red-light setting, mice were placed in the centre and after 1 min testing began. Movement of the animal's body throughout the maze was tracked using ANY-maze tracking software (Stoelting) through an overhead camera for a period of 5 min. The time spent in the open arms was expressed as a percentage of the time spent in both open and closed arms.

**Novelty suppressed feeding.** Before testing, mice were deprived of food for a period of 24 h. During testing, mice were treated with either CNO or vehicle and then placed in a brightly lit open-field arena, which contained ∼5 g of standard chow on top of a 5 cm × 5 cm piece of aluminium foil situated in the arena centre. All mice were consistently placed in the same corner of the arena signalling the start of testing. The time taken to retrieve and begin consuming the food reward in the centre was scored using a stop watch.

**Home cage food consumption.** Before testing, mice were deprived of food for a period of 24 h. During testing, mice were treated with either CNO or vehicle and then given free access to a pre-determined amount of chow (∼5 g) in their home cage. Food intake was recorded 5 min after food was made available. Spillage of food was collected by filter paper placed underneath the chow and was taken into consideration when calculating food intake.

**Hole-board test.** The hole-board apparatus consisted of a clear plexiglass box measuring 50 cm × 50 cm × 50 cm. The floor of the apparatus is a grey wooden platform into which four holes (3 cm in diameter) were cut 10 cm from each corner along the diagonal and the apparatus as a whole was elevated from the ground by 30 cm. Each mouse was tested once following treatment with either CNO or vehicle for a period of 10 min. Once testing began, the animals were allowed to explore the apparatus and the number of head-dips (counted only when ears are below the platform), head-dip time, number of rears, and rearing time were measured. Head dip number and time were scored manually whereas rears and rearing time were scored with ANY-maze (Stoelting) and a side-view camera.

**Alley running for food.** The runway consisted of a 5-cm-wide, 1-m-long platform laterally enclosed by 30 cm black wooden walls. A piece of filter paper with a one-fourth of an M&M mini was placed within 1 cm of the end of the runway. On three consecutive days mice were allowed 10 min to explore the arena and become familiar with the reward placement. On the fourth day food access was restricted. After a 24 h period, mice were then placed at the beginning of the track and the latency to complete the runway to retrieve the food reward was measured using a stopwatch.

**Histology.** Mice were transcardially perfused with PBS, pH 7.4, followed by 4% paraformaldehyde. Brains were extracted and postfixed overnight and subsequently cryoprotected with PBS containing 30% sucrose. Coronal or sagittal 40-μm-thick sections were obtained using a cryostat (Leica, Germany). Sections were slide mounted, counterstained with DAPI (1:500), and imaged using a fluorescence microscope with a ×4.2 objective on an FSX100 fluorescent microscope (Olympus, Japan), or a confocal microscope with a ×20 objective on a Quorum spinning disk confocal microscope (Zeiss, Germany).

**Statistical analyses.** Data were analysed using two- and three-way repeated measures analysis of variances, independent-samples $t$-tests and paired-samples $t$-tests where appropriate. All the data are presented as mean ± s.e., and the difference was considered significant when $P < 0.05$. All experiments were performed by experimenters blind to treatment conditions. All analyses were conducted using Statistica v7.

**Data availability.** Data supporting the findings of this study are available upon request.

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

## Acknowledgements

We thank Alex Levit, Kaori Takehara and Paul Fletcher for technical assistance with behavioural assessment. This research was funded by operating grants to J.C.K. from the Canadian Institutes for Health Research (MOP 496401) and the Natural Sciences and Engineering Council of Canada (NSERC) (MOP 491009). C.J.B. was funded by the doctoral scholarship from NSERC.

## Author contributions

A.J.A., C.J.B. and J.C.K. carried out the study conceptualization and experimental design. A.J.A. and C.J.B. performed and analysed behavioural experiments. D.G., Z.D., C.S.K. acquired and analysed the electrophysiology data. M.A.W. supervised electrophysiology experiments. A.J.A., C.J.B., and J.C.K. wrote the manuscript.

## Additional information

**Competing financial interests:** The authors declare no competing financial interests.

