## [Peer Review File · Nature Communications]

Reviewers' comments:

Reviewer #1 (Remarks to the Author):

Aqrabawi et al present evidence for the role of the AON pars medialis in a series of olfactory behaviors. The role of the AON remains for the most part unknown and this study is important because it is one of the first ones showing a role for mAON in olfactory recognition behavior. The anatomy have precluded the study of mAON function, but the use of new molecular techniques are finally opening the door on the AON function. Using inhibitory chemogenetic tools they showed an increase in the sensitivity to odors, reflected on longer times exploring low concentration odors, reduce time for retreating buried rewards, increased exploration of social cues, compared to empty cages and increased exploration of novel social cues, compared to previously presented social cues. The authors used then an excitatory DREADD construct showing almost (but not exactly!) the mirror image of the results of the inhibitory DREADDs. The authors traced a large projection coming into the mAON from the ventral hippocampus and showed that optogenetic activation of these pathway lead to increase latency for reward retrieval similar to the direct activation of the mAON. These behavioral results seem quite robust and show an important role for the AON in odor recognition.

Main concerns

- 1) The authors speculate that the results might arise by way of the descending input from the mAON into the olfactory bulb, where excitation of the granule cells would inhibit the mitral and tufted cells, reducing the sensitivity of the bulb. Although this interpretation is consistent with the behaviors reported and the anatomy, the fact is that the mAON has projections to other areas, in particular the lateral hypothalamus (Lohman, 1963). Although not as prominent as the connection to the bulb, it might have a disproportionate effect on food related behaviors. The authors need to discuss the alternative possibility that the results observed might be produced by these feedforward projections. Although the mAON does send feedback to the bulb, it should not be presented as the only explanation and other possibilities should be DISCUSSED (not experiments!), given the lack of physiological data during the behavior.
- 2) Same for the role of ventral hippocampus. It does not project only to the mAON, and antidromic optogenetic activation could affect other targets besides the mAON.
- 3) The authors need to do some controls using a GFP virus and CNO injections and repeat the tests in figures 1 and 2. Although the experiments in figure 1 with inhibitory DREADDs are the

mirror of the experiments in figure 2 with excitatory DREADDS, this symmetry is not perfect. In particular figure 1ei shows that CNO treated animals explore more the novel target and less the empty one, which is not mirrored by figure 2di, where we do not see that differences. CNO, a clozapine metabolite might have a systemic effect due to its conversion into clozapine, a well-known antipsychotic, interacting in a complex manner with the effects of the DREADDS on the mAON. Although the controls in Supp. Figure 2 and Supp. Figure 3 help address this concern, it is unclear at which odor concentrations does the manipulation of the AON affects odor exploration.

4) Related to the above point, the slice physiology work on Supp. Figure 1 shows only a modest (~2 mV) membrane depolarization and hyperpolarization with DREADDS and the error bars in panels ii do touch. Given the small number of cells involved (n=5 is the number of cells or animals?), the authors should plot, for each cell, the membrane potential before and after CNO application. Also, the authors should include the data from neuronal excitability, that is the number of action potential as a function of current injected, to give us a better idea of the nature of the change in excitability produced by DREADDS.

Reviewer #2 (Remarks to the Author):

The paper presents a series of experiments that provide some evidence that disruption of mAON is involved in olfactory behaviors such as the duration of head-up sniffing at a Q-tip or social recognition. This is an interesting and novel study that has a potential to offer some of the first insights into the function of an olfactory sensory region. Although these are interesting experiments, several weaknesses make them difficult to interpret.

A major weakness is that all behavioral tests are indirect measurements - investigation time for sensitivity, investigating time for conspecifics as sociability and recognition. While for the sensitivity case it may be somewhat reasonable, for the social measurements the effects can be explained by any change in olfactory processing. Even for the "olfactory sensitivity" tests, sufficient quantification of the behavior and controls (such as 0% odor conc, i.e. pure mineral oil/water conditions) are missing in order to be able to evaluate the claims fully. How variable are these behaviors across animals, and are they differentially impaired by CNO application? Is there a correlation with expression of DREADDs (only histology for 2 animals is shown; localization to mAON in at least one of them is poor)?

It is also not clear why sociability and detection threshold are treated similarly. Why should a change in threshold affect recognition? Crucially none of differences in olfactory behaviors are

convincingly shown to simply result from altered olfactory processing. Perturbations of mAON could affect limbic areas and thus motivation, mood etc. The only control for such general changes is 'locomotion', which seems insufficient.

The hippocampal activation and related the behavioral measurements seem crude and indirect, since many other regions may be involved in the effects (for example, based on Supp Fig 5). The ChR2 experiments in general are not easily interpreted in the current form, as there are serious concerns about off-target stimulation of PFC (along the optical fiber) or striatal regions. Moreover, the number of animals, how they were tested, and how to draw conclusions across experiments that use different DREADDs and used different odor stimuli are unclear.

Since there are no measurements of the effects of the drugs on neuronal activity in vivo, it is hard to know if the effects are due Are the effects due to changes in basal firing or are they specific to odor responses?

Specific Comments:

1. Number of animals used? (and their age); unclear from figure legends if n=11 is 11 mice, or 1 mouse tested 11 times, for example. Also, Figure 1 has n=12 and n=11 for some conditions; why are they not the same? The methods section says each mouse was tested twice; so $6 \text{ mice} * 2 = 12$; why only n=11, if indeed this is how one should think about the n value reported.
2. Figure 1b - black boxes are not easily visible; unclear what i-iv correspond to (presumably boxed regions on left?). Also, Figure 1a appears to be a slightly more anterior slice than their atlas drawing (also please insert coordinates on atlas)?
3. Figure 1c - 2-way ANOVA, why no post-tests? Also, why do the veh animals investigate more with increased concentration? How do you measure they can detect 0.001% odor concentration? I.e. what is the read-out that the mice can even detect these levels? Why is the 0% missing (it is described in the methods section, but not presented in figure 1c).
4. The behavior is insufficiently analyzed and controlled; what is the distribution of sniffing during the sessions? Why only include sniffs with "head up" and 1cm distance to Q-tip (line 296)? Concentration changes with distance to the Q-tip; how did they control for this? What video rate was the data acquired and analyzed at? Was the hand-scoring researcher blinded to the CNO or veh condition? Also, is peanut butter and almond oil equally easily detectable at the concentration ranges tested? Why did they use different odors for the inhibitory vs. excitatory DREADD experiments?

5. How were the parameters for CNO experiments chosen (i.e. 5 mg/kg or 10min waiting // other papers differ). How long did they run the behavior sessions for (i.e. please report, injection time, waited X min, and recorded behavior within X min of injection).
6. Figure 2: (a) doesn't look like mAON? (b) again 0% concentration is missing from the plot.
7. How do the authors measure ability to detect? Investigation time and ability to evoke olfactory responses are not the same measure: "were impaired in their ability to detect the presence of an odour at low concentrations following CNO treatment (Fig. 2b)."
8. ChR2 injection is not projection specific - it is possible that you are also activating surrounding areas such as PFC, or even striatal regions; thus, the results are not interpretable in the present form. Their suppl. Figure 5 also shows the extent of the possible contamination. The fiberoptic can also emit light along the shaft of the fiber (what was the NA and manufacturer of the fiber used as well, this is not reported)
9. Suppl. Figure 1, why low n (n=5) for hM3D?
10. Supl. Figure 3 legend has two "a's"

Reviewer #3 (Remarks to the Author):

In this manuscript, Agrabawi and colleagues report the effects on several olfactory behaviors of bidirectional manipulations of cortical feedback to the mouse main olfactory bulb (MOB). Specifically, the authors manipulate feedback from the medial anterior olfactory nucleus (mAON). They use virally-expressed DREADDs to both inhibit and excite neurons in the mAON during the olfactory behaviors, and then they optogenetically activate a selected input to the mAON that comes from the ventral hippocampus. They see a pattern of results that they interpret as indicating that activity in the mAON generally reduces olfactory sensitivity. They make some interesting observations, but the data feel somewhat preliminary and phenomenological as presented here because there are a number of possibilities for the reason for these effects and the authors do not provide a strong conceptual framework for interpreting what they tell us about the function of this source of cortical feedback to the MOB.

My first major criticism: In my view the authors are not able to make a strong case that altered threshold sensitivity is the explanation for the consequences of mAON perturbation that they observe. All of the behaviors they use convolve threshold sensitivity with other motivational and

behavioral factors that make it difficult to emphatically conclude that their perturbation cleanly alters sensory thresholds. The closest they come to testing threshold sensitivity is the test of investigation of different concentrations of odor, however since they measure spontaneous investigation, it is not possible to dissect the potential contributions of interest level, perceived novelty and motivation. The digging task and social assay are also subject to complex influences of same type and also memory. Moreover, I would argue that the results obtained when activating AON are not inconsistent with simply occluding the sense of smell altogether.

My second major criticism: While there is some internal consistency to the results in that opposing circuit manipulations have apparently opposing behavioral effects, the authors do not offer a conceptual framework for interpreting this pattern and connecting it with other past results. Why and how would AON activity impair odor detection? Through activating granule cells? How do the authors think their manipulation differs from that of Abraham et al (2010, Neuron) who suggested that activating granule cells improved performance on difficult discriminations? The hippocampal input activation experiment is not well motivated and seems arbitrary. Would the results differ if they activated any other excitatory input?

Specific comments:

- 1) What is the investigation time for a cotton swab carrying water? Looking at figures S2a and S3a, it doesn't seem to be very different from many of the data points in figures 1c and 2b. How much investigation do the authors think constitutes detection?
- 2) In the injections into mAON, were labeled terminals observed anywhere in the brain other than the OBs?
- 3) Since the bars in the social assay reciprocally sum to 100%, I gather this is % time investigating with the denominator being total time investigating anything, not total time in the assay. Did absolute investigation time differ between experimental groups?

Please find uploaded our revision of the manuscript NCOMMS-16-06650-T by Aqrabawi et al. entitled “A bidirectional switch for olfaction: top-down modulation of olfactory-guided behaviours by the anterior olfactory nucleus pars medialis and ventral hippocampus”. We are most appreciative of the reviewers’ thoughtful and thorough comments, their overall positive consensus view, and the chance to address each comment and thus improve our submission. Below, we respond point-by-point to review #1, #2, and #3.

Review #1

“Aqrabawi et al present evidence for the role of the AON pars medialis in a series of olfactory behaviors. The role of the AON remains for the most part unknown and **this study is important because it is one of the first ones showing a role for mAON in olfactory recognition behaviour**”.

Point 1) *The authors speculate that the results might arise by way of the descending input from the mAON into the olfactory bulb, where excitation of the granule cells would inhibit the mitral and tufted cells, reducing the sensitivity of the bulb. Although this interpretation is consistent with the behaviors reported and the anatomy, the fact is that the mAON has projections to other areas, in particular the lateral hypothalamus (Lohman, 1963). Although not as prominent as the connection to the bulb, it might have a disproportionate effect on food related behaviors. The authors need to discuss the alternative possibility that the results observed might be produced by these feedforward projections. Although the mAON does send feedback to the bulb, it should not be presented as the only explanation and other possibilities should be DISCUSSED (not experiments!), given the lack of physiological data during the behavior.*

We greatly appreciate the insightful comment and edited the manuscript accordingly to make the scope and limitation of our study more explicit. The mutated versions of human muscarinic receptors, hM4D and hM3D are localized to both neuronal cell bodies and axons, which allows them to be used as anterograde tracer. Intriguingly, we did not observe any significant mCherry-labeled mAON projections in the lateral hypothalamus (LH). However, we acknowledge that expression of these receptors may not serve as a sensitive enough anterograde tracer to visualize the mAON projections in their entirety, including those in the LH. Furthermore, the addition of mCherry as a protein fusion product may also affect the efficient membrane localization of hM3D and 4D. We, therefore, do not rule out the contribution of LH-projecting mAON neurons in the observed behavioural changes, even though they may not be as influential as the AON neurons projecting to the granule cells in the olfactory bulb.

Similar concerns were raised by multiple reviewers and a new series of experiments were conducted. To address whether mAON manipulation changes hunger or food-motivated behaviour, we tested whether CNO treatment would alter home cage food consumption and alley running for a food reward in hM3D and hM4D mice. We found that CNO treatment did not produce any significant change in either of these behaviours, suggesting that mAON manipulation has no overt influence over hunger or motivation to eat.

Point 2) *Same for the role of ventral hippocampus. It does not project only to the mAON, and antidromic optogenetic activation could affect other targets besides the mAON.*

The manuscript now includes new figures showing the effect of activating vHPC inputs to the mPFC on buried food test and social interaction test. Please note that our choice of mPFC was guided by both its heavy connectivity with vHPC as well as its close proximity to the mAON. If there were a major antidromic propagation of action potential spikes, antidromic effects should also be observed in the vHPC terminals in the mPFC. Therefore, selective activation of vHPC terminals in the mPFC would produce similar behavioural changes to those seen upon activating the vHPC terminals in the mAON. The new results show that this is not the case. We found that photostimulation of vHPC terminals in the PFC did not alter latency to uncover a buried food reward, or the ability of mice to distinguish between a novel and familiar conspecific. These results have been added to the manuscript and can be found in Supplementary Figure 7.

Point 3) *The authors need to do some controls using a GFP virus and CNO injections and repeat the tests in figures 1 and 2. Although the experiments in figure 1 with inhibitory DREADDS are the mirror of the experiments in figure 2 with excitatory DREADDS, this symmetry is not perfect. In particular figure 1 shows that CNO treated animals explore more the novel target and less the empty one, which is not mirrored by figure 2di, where we do not see that differences. CNO, a clozapine metabolite might have a systemic effect due to its conversion into clozapine, a well-known antipsychotic, interacting in a complex manner with the effects of the DREADDS on the mAON.*

As the reviewer pointed out, CNO injection alone, especially at high dose (more than 5 mg/kg), may produce behavioural artifacts that are non-specific to the mAON function. In such cases, CNO injection in mice that did not express hM3D or 4D in the mAON would reproduce behavioural changes observed in hM3D and 4D mice. To address this, we analyzed the effect of CNO treatment in animals that received the same surgical procedures as those used in the hM3D and 4D conditions but did not express any detectable levels of the mCherry-DREADD in the AON. We refer to these as a “sham surgery” group. In sham surgery mice, 5 mg/kg CNO treatment had no effect on performance in the buried food test, habituation/dishabituation test, and social interaction tests. These results eliminate the possibility of non-specific behavioural changes mediated by the CNO injection. The results of these experiments can be found in Supplementary Figure 3.

Although the controls in Supp. Figure 2 and Supp. Figure 3 help address this concern, it is unclear at which odor concentrations does the manipulation of the AON affects odor exploration.

In our olfactory sensitivity test, mice were habituated heavily to the cotton swab with mineral oil or water (i.e. 0% odour) in order to minimize novelty responses to the Q-tip or odour vehicle and maximize odour-evoked investigation responses. Habituation trials were significantly longer than test trials, and behaviour was not scored. However, based on the results of the Habituation/Dishabituation tests presented in Supplementary Fig. 2 and 4, we found that CNO treatment did not alter baseline investigation of mineral oil (0% odour) following habituation. Figure 1c and 2b of the manuscript now include insets to make this

point more clear. Thus, hM4D-mediated inhibition of the mAON increased investigation time at 0.001% odour concentration, and hM3D-mediated activation of the mAON reduced investigation time at all three concentrations (0.001%, 0.01%, and 0.1% odour).

Point 4) *The authors should include the data from neuronal excitability, that is the number of action potential as a function of current injected, to give us a better idea of the nature of the change in excitability produced by DREADDS.*

The manuscript now includes new results showing the number of action potential as a function of current injected. We found that CNO increased evoked action potential firing in hM3D-positive neurons and suppressed evoked action potential firing in hM4D-positive neurons. The results of these experiments can be found in Supplementary Figure 1.

Review #2

“The paper presents a series of experiments that provide some evidence that disruption of mAON is involved in olfactory behaviors such as the duration of head-up sniffing at a Q-tip or social recognition. **This is an interesting and novel study that has a potential to offer some of the first insights into the function of an olfactory sensory region**”.

Point 1) *While for the sensitivity case it may be somewhat reasonable, for the social measurements the effects can be explained by any change in olfactory processing. Even for the "olfactory sensitivity" tests, sufficient quantification of the behavior and controls (such as 0% odor conc, i.e. pure mineral oil/or water conditions) are missing in order to be able to evaluate the claims fully.*

We agree with the reviewer. Social recognition is critically dependent on odour detection. It is, however, a complex, multi-step process that can be influenced by other aspects of olfactory processing. While our understanding of the mAON function was shaped by all three tests, it was mainly the olfactory sensitivity threshold test and the buried food test that prompted us to claim for the unique role of the mAON in olfactory sensitivity. Building on the results of the olfactory sensitivity threshold test and the buried food test, we wanted to further verify our main finding using the social interaction test, a behavioural assay highly sensitive to changes in olfaction in mice. We have included a section in the discussion to make the scope and limitation of social interaction test more explicit and discussed the potential contribution of mAON in controlling the olfactory specificity in the context of social interaction.

Regarding the olfactory sensitivity test, the manuscript now includes new graphs showing the investigation times during the habituation session. We found that CNO treatment in hM3D- or hM4D-expressing mice did not alter baseline investigation of mineral oil (0% odour) following habituation. Figure 1c and 2b now include insets to make this point more clear.

How variable are these behaviors across animals, and are they differentially impaired by CNO application? Is there a correlation with expression of DREADDs (only histology for 2 animals is shown; localization to mAON in at least one of them is poor)?

We found that the behavioural phenotypes were very robust and consistent as far as hM3D or 4D expression was properly targeted in the mAON. In addition, we examined a separate group of mice that underwent the same surgical procedures as mice in the hM3D or 4D conditions, but did not express the mCherry-DREADD in the AON. CNO-treatment in these sham control mice did not produce any significant changes in buried food test, habituation/dishabituation test, and social interaction tests. The results of these experiments can be found in Supplementary Figure 3.

Point 2) *It is also not clear why sociability and detection threshold are treated similarly. Why should a change in threshold affect recognition?*

Please see also our response to the reviewer's earlier comment (point 1). Social recognition is a complex, multi-step psychological process, which requires both odour detection and odour discrimination. We speculate that performance in a social recognition task might be sensitive to changes in odour detection, such that the effects of enhanced or reduced olfactory sensitivity to detect social odours may have masked any potential effects of changing olfactory specificity to discriminate social odours. We have added a section in the discussion to elaborate this point.

Crucially none of differences in olfactory behaviors are convincingly shown to simply result from altered olfactory processing. Perturbations of mAON could affect limbic areas and thus motivation, mood etc. The only control for such general changes is 'locomotion', which seems insufficient.

We greatly appreciate the insightful comment. As the reviewer rightly pointed out, performances in any behavioural tasks requiring voluntary actions can potentially be influenced by changes in other psychological processes such as anxiety, general exploration, and motivation. To address these concerns, which were raised by multiple reviewers, we have added control experiments to test whether manipulating mAON activity could produce changes in other behavioural measures. We tested new groups of mice expressing hM4D or hM3D in the mAON for anxiety-like behaviour (elevated plus maze and novelty suppressed feeding test), general exploratory behaviour (hole-board test), and motivation for food reward (runway test). Our new data indicate that CNO-treated hM3D or hM4D mice do not show any significant changes in anxiety-like behaviour, general exploratory behaviour, or motivation. The results of these experiments can be found in Figure 3.

Point 3) *The hippocampal activation and related the behavioral measurements seem crude and indirect, since many other regions may be involved in the effects (for example, based on Supp Fig 5). The Chr2 experiments in general are not easily interpreted in the current form, as there are serious concerns about off-target stimulation of PFC (along the optical fiber) or striatal regions.*

To address some of the concerns raised by the reviewer, we have included a control experiment wherein performance of the buried food test and social interaction test were repeated during stimulation of the vHPC-PFC pathway. Here, mice received bilateral vHPC infusions of ChR2-YFP and bilateral implantation of optical fibers targeting the PFC. Mice underwent identical testing procedures using identical laser stimulation parameters. Photostimulation of vHPC terminals in the PFC did not alter latency to uncover a buried food reward, or the ability of mice to distinguish between a novel and familiar conspecific. These results have been added to the manuscript and can be found in Supplementary Figure 7.

Moreover, the number of animals, how they were tested, and how to draw conclusions across experiments that use different DREADDs and used different odor stimuli are unclear.

Thanks for your careful attention. We have addressed these concerns in our revised manuscript. Please see also our responses to specific comments below.

Since there are no measurements of the effects of the drugs on neuronal activity in vivo, it is hard to know if the effects are due to changes in basal firing or are they specific to odor responses?

The reviewer raises an important consideration that would provide useful clarification of the results, particularly in terms of temporal dynamics of mAON activity. While the revised manuscript does not offer *in vivo* electrophysiology data to show the CNO effects in freely-behaving mice, we have added slice electrophysiology data to address the effect of CNO on the excitability of mAON neurons *in vitro* (Supplementary Figure 1). We are working towards *in vivo* recording of the mAON neuron firing, but we feel that the experiment to reliably measure the odour specific responses of the mAON neurons *in vivo* is very time-consuming and out of the scope for the current manuscript, which mainly focuses on the behavioural contribution of mAON activity.

Point 4) Specific comments

*1. Number of animals used? (and their age); unclear from figure legends if n=11 is 11 mice, or 1 mouse tested 11 times, for example. Also, Figure 1 has n=12 and n=11 for some conditions; why are they not the same? The methods section says each mouse was tested twice; so 6 mice*2 = 12; why only n=11, if indeed this is how one should think about the n value reported.*

The 'n' refers to the number of animals used during behavioural testing. For each hM4D and hM3D groups, 12 mice were prepared (i.e. total of 24 mice expressing either DREADD). For all behavioural paradigms, with the exception of the social interaction test (and the additional control experiments included in the revision of this manuscript), we adopted a within-subjects design such that each mouse was tested twice – following treatment with either CNO or vehicle in a counterbalanced fashion. However, successful performance in the social recognition phase of the social interaction test appeared to depend heavily on novelty, and repetition of the task, which is required for a within-subjects design, was not feasible in our hands. Thus, an additional 12 mice per group (hM4D or hM3D) were prepared to increase the n-value to 12 for each treatment condition

(CNO or vehicle). In summary, a total of 24 mice were tested in all assays (12 hM4D and 12 hM3D) but an additional 24 were tested in the social interaction test alone (24 hM4D and 24 hM3D). Where the n-value is lower than 12 is a reflection of the number of animals used to generate figures and perform statistical analyses after the removal of outliers.

2. *Figure 1b - black boxes are not easily visible; unclear what i-iv correspond to (presumably boxed regions on left?). Also, Figure 1a appears to be a slightly more anterior slice than their atlas drawing (also please insert coordinates on atlas)?*

The proper adjustments to improve the clarity of this figure have been made.

3. *Figure 1c - 2-way ANOVA, why no post-tests? Also, why do the veh animals investigate more with increased concentration? How do you measure they can detect 0.001% odor concentration? I.e. what is the read-out that the mice can even detect these levels? Why is the 0% missing (it is described in the methods section, but not presented in figure 1c).*

Post-hoc tests were unnecessary as the overall 2-way ANOVA describes a change in investigation time as a function of both group and concentration. This most accurately and stringently describes the trend in the data used for our interpretation.

The olfactory sensitivity task employed here assesses whether an animal can detect an odour based on the ability of an odour to evoke approach and investigation. The odour was novel and mildly appetitive – peanut butter or almond. Animals naturally investigate novel odours, particularly if they are appetitive (see habituation dishabituation, particularly female odour). Thus, the odours presented should elicit investigation if the animal can detect the presence of the odour. If the odour concentration does not elicit investigation, it is likely due to an inability to detect the odour. Odour concentrations were chosen to capture a very low concentration that may not be detected normally, and were based on Soria-Gomez et al. 2014, Nature Neuroscience.

Increases in investigation may be due to either a higher proportion of mice detecting and investigating the odour, the 10-fold increase in ease of detection, or the appetitive nature of the odour stimulus.

Animals were heavily habituated to the vehicle for the odour stimuli on the Q-tip to ensure that investigation times in future trials were not contaminated with novelty responses to the Q-tip or the 0% vehicle. As a result, this habituation trial was significantly longer in duration than other trials, and investigation time was not scored. However, based on the results of the Habituation/Dishabituation tests presented in Supplementary Fig. 2 and 4, we found that CNO treatment did not alter baseline investigation of mineral oil (0% odour) following habituation. Figures 1c and 2b of the manuscript now include insets to clarify this point.

4. *The behavior is insufficiently analyzed and controlled; what is the distribution of sniffing during the sessions? Why only include sniffs with "head up" and 1cm distance to Q-tip (line 296)? Concentration changes with distance to the Q-tip; how did they control for this? What video rate was the data acquired and analyzed at? Was the hand-scoring researcher blinded to the CNO or veh condition?*

“Head up” sniffing is a qualitative term used to describe goal-directed, investigatory sniffing patterns. While concentration does change with distance from the Q-tip, this is difficult to quantify, and attempts to control for this would likely increase the error of measurement in the data. The parameters used to determine what constitutes an “investigation” were chosen to provide a very conservative approximation. If the animal positions their sniffing away from the Q-tip at a distance greater than 1 cm, one cannot reliably determine that the Q-tip is the target of interest. A video was captured for each session at a frame rate of 60 fps using a NIKON D5200 camera equipped with a 35 mm prime lens so that an experimenter blind to the treatment was able to score the videos frame-by-frame to determine more accurately the duration of investigation. This information was added to the methods.

Also, is peanut butter and almond oil equally easily detectable at the concentration ranges tested? Why did they use different odors for the inhibitory vs. excitatory DREADD experiments?

A single stock solution of both the peanut butter and almond extract was used for hM4D and hM3D experiments to control for any variation in concentration during preparation. Both groups were tested with 7.5 μ L of both odours in a counter-balanced manner to control for any effect that may be specific to a single odour. We have updated the method section accordingly.

5. How were the parameters for CNO experiments chosen (i.e. 5 mg/kg or 10min waiting // other papers differ). How long did they run the behavior sessions for (i.e. please report, injection time, waited X min, and recorded behavior within X min of injection).

These procedures are described in the supplementary methods. The protocol is based on studies by our lab and others which have successfully applied the DREADDs technique to produce behavioural changes. A high concentration of CNO and a waiting period were particularly important to ensure binding of the ligand to the receptor. Behavioural testing sessions differed in terms of their duration, but none were longer than 30 minutes in total (i.e. testing was completed 40 minutes following CNO treatment).

6. Figure 2: (a) doesn't look like mAON? (b) again 0% concentration is missing from the plot.

Figure 2a represents a more posterior section of the mAON, but still well within the boundaries of the structure. Point 6b) was addressed in response to point 3.

7. How do the authors measure ability to detect? Investigation time and ability to evoke olfactory responses are not the same measure: "were impaired in their ability to detect the presence of an odour at low concentrations following CNO treatment (Fig. 2b)."

Odour detection can be measured at both neuronal and behavioural level, depending on how one defines it. As discussed in point 3, the olfactory sensitivity test capitalized on the ability of an odour, once detected, to elicit behavioural responses (e.g. approach and investigation). In our opinion, measuring such odour-evoked behavioural response is a more stringent, if not more sensitive, way to evaluate the ability to detect odour, compared

to a measure of odour-evoked neuronal responses (e.g. electrophysiological responses). That is because behavioural responses require the detection of odour at neuronal level, whereas the presence of neural representation of an odour does not necessarily accompany detectable behavioural responses - especially when the odour stimulus is weak. This distinction between neural versus behavioural responses is an important topic to consider when interpreting the effects of mAON manipulation, but we feel that experimental demonstration of the distinction may fall out of the scope of the current manuscript.

Again, we fully acknowledge that the absence of behavioural responses does not always indicate the absence of odour detection at neuronal level. Animals capable of detecting odours may not engage in approach and investigation for various reasons including changes in motivation and anxiety. Our new data (Fig. 3) indicate that CNO-treated hM3D or hM4D mice do not show any significant changes in anxiety-like behaviour, general exploratory behaviour, or motivation for food reward. Thus, observed increase or decrease in time spent investigating odour is likely due to changes in ability to detect odour.

8. *ChR2 injection is not projection specific - it is possible that you are also activating surrounding areas such as PFC, or even striatal regions; thus, the results are not interpretable in the present form. Their suppl. Figure 5 also shows the extent of the possible contamination. The fiberoptic can also emit light along the shaft of the fiber (what was the NA and manufacturer of the fiber used as well, this is not reported)*

We have included a control experiment to address the possibility that the effects of vHPC-mAON pathway activation may be mediated by PFC responses through light contamination or due to back propagation to the vHPC. Activation of the vHPC-PFC pathway had no effect on performance in the buried food test or social recognition test. More methodological details regarding the optogenetic approach have been added to the supplementary methods.

9. *Suppl. Figure 1, why low n (n=5) for hM3D?*

The purpose of this experiment is to demonstrate the ability of hM3D to depolarize the membrane potential. Only five cells were necessary to observe this difference clearly.

10. *Supl. Figure 3 legend has two "a's"*

Thank you, this has now been fixed.

Reviewer #3

“They see a pattern of results that they interpret as indicating that activity in the mAON generally reduces olfactory sensitivity...**there are a number of possibilities for the reason for these effects** and the authors do not provide a strong conceptual framework for interpreting what they tell us about the function of this source of cortical feedback to the MOB”.

Point 1) *In my view the authors are not able to make a strong case that altered threshold sensitivity is the explanation for the consequences of mAON perturbation that they observe. All*

of the behaviors they use convolve threshold sensitivity with other motivational and behavioral factors that make it difficult to emphatically conclude that their perturbation cleanly alters sensory thresholds. The closest they come to testing threshold sensitivity is the test of investigation of different concentrations of odor, however since they measure spontaneous investigation, it is not possible to dissect the potential contributions of interest level, perceived novelty and motivation. The digging task and social assay are also subject to complex influences of same type and also memory.

We appreciate the insightful comments. While the olfactory sensitivity test measures the threshold for odour detection as an index of odour sensitivity, the total time spent exploring each dilution can be influenced by other factors including anxiety, general exploration, and motivation, confounding obtained results. However, this is somewhat an intrinsic limitation of all rodent olfactory tests involving voluntary actions. Thus, one can never completely rule out a potential contribution of non-olfactory factors in rodent olfactory tests. Nevertheless, we agree that it is critical to identify any major changes in other behavioural measures. To address this, we tested new groups of mice expressing hM4D or hM3D in the mAON for anxiety-like behaviour (elevated plus maze and novelty suppressed feeding test), general exploratory behaviour (hole-board test), and motivation for food reward (runway test). We observed no changes in any of these tests following CNO treatment, suggesting that manipulating mAON activity does not produce any significant changes in anxiety-like behaviour, general exploration, or motivation. The results of these experiments can be found in Figure 3.

Moreover, I would argue that the results obtained when activating AON are not inconsistent with simply occluding the sense of smell altogether.

We found that activating the mAON reduces, but not eliminates, the sense of smell because animals were capable of completing olfactory-dependent tasks such as the buried food test (albeit significantly more slowly) and the habituation/dishabituation test. However, we agree with the reviewer that it is still a matter of debate whether the AON cortical feedback signals simply provide a nonspecific global inhibitory gain control over mitral/tuft cells across all glomeruli, or they can trigger cell- or odour-specific changes, actively shaping odour representation in the OB. We feel that this is a very important question that should be addressed by follow-up studies.

Point 2) *While there is some internal consistency to the results in that opposing circuit manipulations have apparently opposing behavioral effects, the authors do not offer a conceptual framework for interpreting this pattern and connecting it with other past results. Why and how would AON activity impair odor detection? Through activating granule cells? How do the authors think their manipulation differs from that of Abraham et al (2010, Neuron) who suggested that activating granule cells improved performance on difficult discriminations?*

Thanks for pointing out this very important contrast between our results and the Abraham et al study. Both studies activated granule cells; ours indirectly by activating cortical inputs arriving at granule cells and Abraham et al. by the direct activation of granule cells in the olfactory bulb. However, the two studies employed different olfactory tests. Our study

focused on the effect of manipulating AON cortical inputs to granule cells on olfactory sensitivity and general olfaction-guided behaviours. In contrast, Abraham et al focused on the effect of granule cell activation on more difficult odour discrimination learning.

Reconciling the findings from these two studies provides a very important conceptual framework: enhancing the ability to discriminate similar odours (e.g. by activating granule cells) comes at a cost of lowering olfactory sensitivity to detect an odour. Accordingly, enhancing olfactory sensitivity (e.g. by inhibiting the mAON) would facilitate odour detection but likely compromise animal's ability to discriminate similar odours by introducing more noise to neural representation of odour signals at the olfactory bulb. The mAON has been shown to activate granule cells in the bulb and evoke inhibitory postsynaptic responses in the mitral cells in vivo. Thus, we predict that hM3D-mediated mAON activation would lead to enhancements in odour discrimination as observed in Abraham et al. The revised manuscript now includes this conceptual framework and relevant discussion of the related topics.

The hippocampal input activation experiment is not well motivated and seems arbitrary.

The finding that the mAON controls the olfactory sensitivity in a bidirectional manner prompted us to investigate which higher-order limbic structures can directly change the mAON activity. Specifically, we reasoned that such dynamic modulation of olfactory sensitivity is likely modified by limbic inputs regarding emotional or motivational states. The necessity of testing the role of vHPC-mAON projection became only apparent as the retrograde tracing experiment revealed the vHPC as a primary source of afferents to the mAON.

Would the results differ if they activated any other excitatory input?

It is indeed plausible that other inputs to the mAON can produce different effects on olfactory behaviours. For example, oxytocin and serotonin inputs are two major neuromodulatory signals that can potentially influence the mAON function in vivo. While we appreciate the reviewer's insight into this important possibility, we feel that experiment to address this question is out of the scope for the current manuscript.

Point 3) Specific comments

1. *What is the investigation time for a cotton swab carrying water? Looking at figures S2a and S3a, it doesn't seem to be very different from many of the data points in figures 1c and 2b. How much investigation do the authors think constitutes detection?*

Our definition of detection was a significant increase in investigation time across increasing concentrations of the odour, and a change in sensitivity produced by CNO was defined as a difference between vehicle treatment and CNO treatment.

2. *In the injections into mAON, were labeled terminals observed anywhere in the brain other than the OBs?*

mCherry-labeled signals were observed only in the OB.

3. *Since the bars in the social assay reciprocally sum to 100%, I gather this is % time investigating with the denominator being total time investigating anything, not total time in the assay. Did absolute investigation time differ between experimental groups?*

The reviewer is correct, and the denominator used was total time investigating either the new or old conspecific. These % results are in fact driven by differences in absolute investigation times. Where proportional investigation times differed between groups, we have also added statistics comparing absolute investigation times in the figure captions.

Reviewers' Comments:

Reviewer #1 (Remarks to the Author):

The authors have successfully addressed most of my concerns. Adding the remaining points to the discussion will make for a very nice paper. However, in terms of presentation, the authors use bar graphs with error bars. For most of the manipulations, it is a paired comparison of with/without CNO/light (and indeed the authors used paired t-tests, which is the right test). The authors should present individual data points with and without manipulation joined by a line, in order for the readers to evaluate the diversity in responses.

“Aqrabawi et al present evidence for the role of the AON pars medialis in a series of olfactory behaviors. The role of the AON remains for the most part unknown and this study is important because it is one of the first ones showing a role for mAON in olfactory recognition behaviour”.

Point 1) The authors speculate that the results might arise by way of the descending input from the mAON into the olfactory bulb, where excitation of the granule cells would inhibit the mitral and tufted cells, reducing the sensitivity of the bulb. Although this interpretation is consistent with the behaviors reported and the anatomy, the fact is that the mAON has projections to other areas, in particular the lateral hypothalamus (Lohman, 1963). Although not as prominent as the connection to the bulb, it might have a disproportionate effect on food related behaviors. The authors need to discuss the alternative possibility that the results observed might be produced by these feedforward projections. Although the mAON does send feedback to the bulb, it should not be presented as the only explanation and other possibilities should be DISCUSSED (not experiments!), given the lack of physiological data during the behavior.

We greatly appreciate the insightful comment and edited the manuscript accordingly to make the scope and limitation of our study more explicit. The mutated versions of human muscarinic receptors, hM4D and hM3D are localized to both neuronal cell bodies and axons, which allows them to be used as anterograde tracer. Intriguingly, we did not observe any significant mCherry-labeled mAON projections in the lateral hypothalamus (LH). However, we acknowledge that expression of these receptors may not serve as a sensitive enough anterograde tracer to visualize the mAON projections in their entirety, including those in the LH. Furthermore, the addition of mCherry as a protein fusion product may also affect the efficient membrane localization of hM3D and 4D. We, therefore, do not rule out the contribution of LH-projecting mAON neurons in the observed behavioural changes, even though they may not be as influential as the AON neurons projecting to the granule cells in the olfactory bulb.

Similar concerns were raised by multiple reviewers and a new series of experiments were conducted. To address whether mAON manipulation changes hunger or food-motivated behaviour, we tested whether CNO treatment would alter home cage food consumption and alley running for a food reward in hM3D and hM4D mice. We found that CNO treatment did not produce any significant change in either of these behaviours, suggesting that mAON manipulation has no overt influence over hunger or motivation to eat.

The authors need to include in the discussion the possibility that the results could be result of projections of the AON to other areas.

Point 2) Same for the role of ventral hippocampus. It does not project only to the mAON, and antidromic optogenetic activation could affect other targets besides the mAON.

The manuscript now includes new figures showing the effect of activating vHPC inputs to the mPFC on buried food test and social interaction test. Please note that our choice of mPFC was guided by both its heavy connectivity with vHPC as well as its close proximity to the mAON. If there were a major antidromic propagation of action potential spikes, antidromic effects should also be observed in the vHPC terminals in the mPFC. Therefore, selective activation of vHPC terminals in the mPFC would produce similar behavioural changes to those seen upon activating the vHPC terminals in the mAON. The new results show that this is not the case. We found that photostimulation of vHPC terminals in the PFC did not alter latency to uncover a buried food reward, or the ability of mice to distinguish between a novel and familiar conspecific. These results have been added to the manuscript and can be found in Supplementary Figure 7.

The authors have addressed this concern with new experiment. However, the injection into the ventral hippocampus in figure 4a seems very widespread and there seem to be some fluorescence into other areas like the amygdala. The authors should include pictures showing that there are no cell bodies there, as these other areas might project to the AON or the bulb. It is not clear from the current picture if there are cell bodies outside the ventral hippocampus.

The authors should also include a higher resolution image for figure 2bi, there seem to be axons in the glomerular layers and their presence should be discussed.

Point 3) The authors need to do some controls using a GFP virus and CNO injections and repeat the tests in figures 1 and 2. Although the experiments in figure 1 with inhibitory DREADDS are the mirror of the experiments in figure 2 with excitatory DREADDS, this symmetry is not perfect. In particular figure 1 shows that CNO treated animals explore more the novel target and less the empty one, which is not mirrored by figure 2di, where we do not see that differences. CNO, a clozapine metabolite might have a systemic effect due to its conversion into clozapine, a well-known antipsychotic, interacting in a complex manner with the effects of the DREADDS on

the mAON.

As the reviewer pointed out, CNO injection alone, especially at high dose (more than 5 mg/kg), may produce behavioural artifacts that are non-specific to the mAON function. In such cases, CNO injection in mice that did not express hM3D or 4D in the mAON would reproduce behavioural changes observed in hM3D and 4D mice. To address this, we analyzed the effect of CNO treatment in animals that received the same surgical procedures as those used in the hM3D and 4D conditions but did not express any detectable levels of the mCherry-DREADD in the AON. We refer to these as a “sham surgery” group. In sham surgery mice, 5 mg/kg CNO treatment had no effect on performance in the buried food test, habituation/dishabituation test, and social interaction tests. These results eliminate the possibility of non-specific behavioural changes mediated by the CNO injection. The results of these experiments can be found in Supplementary Figure 3.

The authors still need to discuss the difference between figures 1e and 2d.

Although the controls in Supp. Figure 2 and Supp. Figure 3 help address this concern, it is unclear at which odor concentrations does the manipulation of the AON affects odor exploration.

In our olfactory sensitivity test, mice were habituated heavily to the cotton swab with mineral oil or water (i.e. 0% odour) in order to minimize novelty responses to the Q-tip or odour vehicle and maximize odour-evoked investigation responses. Habituation trials were significantly longer than test trials, and behaviour was not scored. However, based on the results of the Habituation/Dishabituation tests presented in Supplementary Fig. 2 and 4, we found that CNO treatment did not alter baseline investigation of mineral oil (0% odour) following habituation. Figure 1c and 2b of the manuscript now include insets to make this point more clear. Thus, hM4D-mediated inhibition of the mAON increased investigation time at 0.001% odour concentration, and hM3D-mediated activation of the mAON reduced investigation time at all three concentrations (0.001%, 0.01%, and 0.1% odour).

The authors have successfully addressed this concern.

Point 4) The authors should include the data from neuronal excitability, that is the number of action potential as a function of current injected, to give us a better idea of the nature of the change in excitability produced by DREADDs.

The manuscript now includes new results showing the number of action potential as a function of current injected. We found that CNO increased evoked action potential firing in hM3D-positive neurons and suppressed evoked action potential firing in hM4D-positive neurons. The results of

these experiments can be found in Supplementary Figure 1.

The authors have successfully addressed this concern.

Reviewer #2 (Remarks to the Author):

The authors have responded to the comments mainly with arguments, and a few additional (control) experiments. I continue to have some conceptual issues (apparently shared by reviewer 3 at least), but I don't have additional concerns to hold up publication. I hope the post-publication review by others in this research field will act to judge this work in an objective manner.

Reviewer #3 (Remarks to the Author):

Overall I think the authors have done a reasonably good job of responding to my critiques. Particularly in response to my two most important concerns:

1) Doubt about the specificity of the behavior effect to olfaction.

In response to this concern raised by all reviewers, the authors perform a number of behavioral controls to show that their manipulation doesn't affect readouts of potentially confounding influences. Ultimately, this is all negative evidence, but the authors are correct that it is difficult to exclude those confounds when measuring olfactory function with volitional responses.

2) Lack of a conceptual framework and integration with the literature.

The authors have now done a more thoughtful job of considering the significance of the results in context with the literature.

In light of these improvements, I am more or less ok with the manuscript, with the following small exceptions:

 It is unfortunate that the authors weren't able to show that the behavioral controls that they added were still capable of showing the effects of mAON manipulation on olfactory behavior. At the very least, they should quantitatively compare the extent of labeling (number of cells, area of infection, etc.) between the original set of mice and the new controls. The concern of course is if the controls had consistently weaker CNO effect.

 I still think the hippocampus activation only shows that it likely *can* alter olfactory processing, but doesn't prove that it actually does under normal conditions. I would be satisfied if the manuscript were a bit more explicit about this limitation. They aren't far off, but I think they should be clear that ventral hippocampus is at this point merely a candidate influence acting via mAON

Please find uploaded our revision of the manuscript NCOMMS-16-06650A by Agrabawi et al. entitled “A *bidirectional switch for olfaction: top-down modulation of olfactory-guided behaviours by the anterior olfactory nucleus pars medialis and ventral hippocampus*”. We are most appreciative of the reviewers’ thoughtful and thorough comments, their overall positive consensus view, and the chance to address each comment and thus improve our submission. Below, we respond point-by-point to review #1, #2, and #3.

Reviewer #1: The authors have successfully addressed most of my concerns. Adding the remaining points to the discussion will make for a very nice paper. However, in terms of presentation, the authors use bar graphs with error bars. For most of the manipulations, it is a paired comparison of with/without CNO/light (and indeed the authors used paired t-tests, which is the right test). *The authors should present individual data points with and without manipulation joined by a line, in order for the readers to evaluate the diversity in responses.*

Figures 1d and 2c depicting the results of the buried food test, which was a within-subjects manipulation, are now presented as individual points connected by a line. We have also added two separate panels to supplementary figure 9 for the pre- and post-stimulation phases of the buried food test conducted in vHPC-mAON mice depicting the data from Figure 4c in this way.

Point 1) *The authors need to include in the discussion the possibility that the results could be result of projections of the AON to other areas.*

We have included the following paragraph in the discussion to acknowledge the possibility.

“It is possible that mAON projections to regions other than the OB can explain some of the behavioural effects observed in these experiments. For example, projections to the hypothalamus may alter food-related behaviours and affect performance of the BFT, while projections to the amygdala may promote anxious states that disrupt behavioural performance in general.”

Point 2) *However, the injection into the ventral hippocampus in figure 4a seems very widespread and there seem to be some fluorescence into other areas like the amygdala. The authors should include pictures showing that there are no cell bodies there, as these other areas might project to the AON or the bulb. It is not clear from the current picture if there are cell bodies outside the ventral hippocampus. The authors should also include a higher resolution image for figure 2bi, there seem to be axons in the glomerular layers and their presence should be discussed.*

We have included confocal images showing that there are very few, if any, ChR2-labelled cell bodies in the amygdala areas immediately lateral to the infected area of the vHPC. In addition, a higher resolution image has been added in the Figure 1 to show that mCherry-positive mAON fibers are present in the glomerular layers, although they are scarce.

Point 3) *The authors still need to discuss the difference between figures 1e and 2d.*

We have included the following sentences in the manuscript to discuss this difference.

“In the social interaction test, mAON activation impaired social recognition as mice treated with CNO spent an approximately equal proportion of time investigating a novel and familiar conspecific (Fig. 2d-ii), supporting the role of the mAON in modulating olfactory sensitivity. However, unlike mAON inhibition, activation of the structure produced no change in investigation time during the sociability phase (Fig. 2d-i), when a stranger conspecific is present only in one side of the arena, while the other side is left empty. It is likely that despite their reduced olfactory sensitivity, mice relied on other sensory (e.g. visual) stimuli to guide their investigation of the individual conspecific.

Reviewer #2: The authors have responded to the comments mainly with arguments, and a few additional (control) experiments. I continue to have some conceptual issues (apparently shared by reviewer 3 at least), but **I don't have additional concerns to hold up publication.** I hope the post-publication review by others in this research field will act to judge this work in an objective manner.

Thank you for your comments.

Reviewer #3: Overall I think the authors have done a reasonably good job of responding to my critiques. Particularly in response to my two most important concerns: 1) Doubt about the specificity of the behavior effect to olfaction. In response to this concern raised by all reviewers, the authors perform a number of behavioral controls to show that their manipulation doesn't affect readouts of potentially confounding influences. Ultimately, this is all negative evidence, but the authors are correct that it is difficult to exclude those confounds when measuring olfactory function with volitional responses. 2) Lack of a conceptual framework and integration with the literature. The authors have now done a more thoughtful job of considering the significance of the results in context with the literature. **In light of these improvements, I am more or less ok with the manuscript, with the following small exceptions:**

Point 1) *It is unfortunate that the authors weren't able to show that the behavioral controls that they added were still capable of showing the effects of mAON manipulation on olfactory behavior. At the very least, they should quantitatively compare the extent of labeling (number of cells, area of infection, etc.) between the original set of mice and the new controls. The concern of course is if the controls had consistently weaker CNO effect.*

We greatly appreciate this insightful comment. In anticipation of reviewer comments on our previous submission, we kept the mice used for the control experiments to perform any additional experiments on. Thus, we retested these mice in the buried food test to address the point raised by the reviewer. We have now included data confirming that the new behavioural control group could still show the effects of mAON manipulation in the buried food test to the same extent found in the original set of mice, which are shown in Supplementary Fig. 5. This suggests that the lack of behavioural changes in motivation, anxiety, exploration, or motivation in the new control group were not due to the weak CNO effect.

Point 2) *I still think the hippocampus activation only shows that it likely *can* alter olfactory processing, but doesn't prove that it actually does under normal conditions. I would be satisfied if the manuscript were a bit more explicit about this limitation. They aren't far off, but I think*

they should be clear that ventral hippocampus is at this point merely a candidate influence acting via mAON

We agree with the reviewer that we demonstrate a candidate region for influencing AON activity. We have altered the manuscript to reflect this more conservative approach by adding the following statement:

“While the present study suggests that the vHPC is a candidate region for higher-order modulation of olfactory sensitivity via the mAON.” We also now emphasize in the discussion that, “future studies should identify the functional significance of other neocortical and limbic structures positioned to alter olfaction through the mAON”

Reviewers' Comments:

Reviewer #1 (Remarks to the Author):

The manuscript is ready for publication.

Reviewer #3 (Remarks to the Author):

The authors' response is OK with me